# mSOP-765k: A Benchmark For Multi-Modal Structured Output Predictions

**Bianca Lamm**  *Bianca.Lamm@de.markant.com*
*Markant Services International GmbH*

**Janis Keuper**  *keuper@imla.ai*
*IMLA, Offenburg University and University of Mannheim*

**Reviewed on OpenReview:** *https://openreview.net/forum?id=H7eYL4yFZS*

## Abstract

This paper introduces *mSOP-765k*, a large-scale benchmark for the evaluation of *multi-modal Structured Output Prediction* (mSOP) pipelines. Besides novel evaluation metrics, the benchmark provides combined training and test datasets with over 765,000 images taken from real-world product advertisements. Each of these images contains product visualizations, textual information like product name or brand, and numerical data such as product weight, price, and discount. All images are annotated with the corresponding structured information in form of dictionaries containing key-value pairs.

An initial baseline evaluation, including various LLMs and VLMs, as well as multi-modal RAG approaches, shows that the proposed benchmark provides a challenging problem which can not yet be solved completely by state-of-the-art mSOP methods. The benchmark and dataset are available under a creative-commons license: https://www.msop-765k.org/.

## 1 Introduction

Large Language Models (LLMs) (Vaswani et al.; Zhao et al.) have become indispensable components in modern natural language processing, demonstrating remarkable capabilities across a wide range of tasks. More recently, Vision Language Models (VLMs) (Radford et al.; Zhang et al.) have extended this paradigm by enabling the processing and understanding of multi-modal inputs, effectively bridging the gap between visual and textual information. Although both LLMs and VLMs primarily generate responses in human-readable text, numerous applications require outputs in machine-readable formats. Addressing this need, recent advances have endowed LLMs and VLMs with the ability to produce Structured Output Predictions (SOPs) (Liu et al., c), allowing their predictions to be provided as well-defined and easily parseable data representations. This emerging capability opens new avenues for leveraging these models in complex workflows that require precise and unambiguous interpretation of generated content.

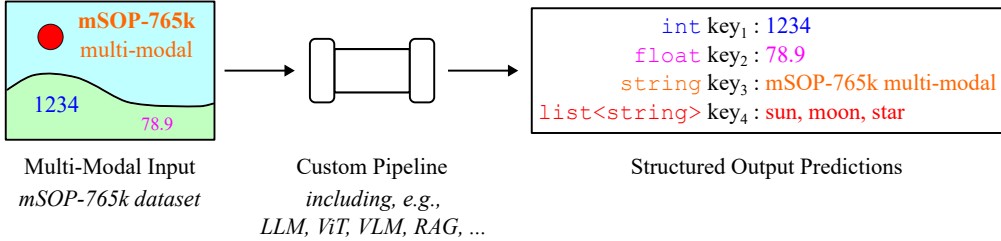

Figure 1: Illustration of the multi-modal Structured Output Prediction task performed with the *mSOP-765k* dataset. The proposed benchmark measures the ability of a given processing pipeline to predict structured key-value pairs of textual, numerical, and list data.

A comprehensive benchmark with clearly defined quality metrics for the systematic evaluation of multi-modal SOPs is essential to advance the evolving research and the practical application of multi-modal structured output prediction methods. However, to the best of our knowledge, so far no large-scale benchmark exists that thoroughly assesses these capabilities across diverse structured prediction tasks on multiple modalities. In order to fill this gap, we introduce *mSOP-765k*. Figure 1 illustrates the evaluation of provided multi-modal structured output prediction pipelines, such as LLMs (Vaswani et al.; Zhao et al.), VLMs Radford et al.; Zhang et al., Vision Transformers (ViTs) (Dosovitskiy et al.), or Retrieval Augmented Generation (RAG) (Gao et al.; Lamm & Keuper, b) systems by the proposed benchmark.

An extensive baseline evaluation of current approaches shows, that while many methods are able to predict single key-value pairs with high accuracy, all tested algorithms essentially fail to predict consistent multi-value data structures in a more complex scenario.

**The following key contributions are presented in this paper:**

- We introduce **mSOP-765k**, the first large scale benchmark for structured output predictions from multi-modal inputs. *mSOP-765k* provides over 765 thousand images of product advertisements containing product visualizations, textual product descriptions and promotions, as well as numerical data like prices and discounts. Each image has been manually annotated with detailed key-value pairs provided in a machine readable data-structure. The benchmark and dataset are available under a creative-commons license.

- The benchmark introduces novel evaluation metrics, allowing to measure the prediction quality of complex output structures.

- A first baseline evaluation of a wide range of different approaches, including LLMs (Vaswani et al.; Zhao et al.), VLMs Radford et al.; Zhang et al., ViTs (Dosovitskiy et al.), or RAG (Gao et al.; Lamm & Keuper, b) systems on *mSOP-765k* clearly shows the strong limitations of current approaches towards accurate predictions of complex structured output from multi-modal inputs.

## 2 Related Work

### 2.1 Creation of Structured Output Predictions

Most recent LLMs and VLMs are able to create a machine readable output from natural language or visual inputs (Wang et al.). Their implementations, as far as publicly known, are based on several different approaches to generate and / or improve SOPs which have been proposed in literature.

Irugalbandara introduces *Meaning Typed Prompting (MTP)*, which integrates the definitions of types, meanings, and abstractions in the prompting process. The authors evaluate the performance of their approach on the text based *LLM Structured Output Benchmark* (Leo). The paper also includes a small case study that analyzes the generation of SOPs from a single input image using GPT-4o-mini (Irugalbandara).

The *Generate and Organize (G&O)* method has been introduced by Li et al.. This approach improves the capability of LLMs to produce SOPs in a two step process which separates the content generation and structuring: first, the response is generated in natural language and then a second LLMs call is used to organize and format the received response into the intended structure. Li et al. focus their evaluation on the tasks of *Named Entity Recognition* and *Relation Extraction*, which exclusively process textual input. Consequently, the models investigated in this study are limited to LLMs.

Wang et al. present *SLOT*, an abbreviation for Structured LLM Output Transformer. This approach converts the unstructured responses of LLM into structured formats by applying a fine-tuned lightweight language model as a post-processing layer. The evaluation of *SLOT* in (Wang et al.) also exclusively investigates text-to-structure tasks.

Lamm & Keuper (b) propose the *Visual RAG Pipeline*, which generates SOPs given multi-modal inputs comprising both image and text data. Their evaluation includes the analysis of various commercial models, specifically GPT-4o, GPT-4o-mini, and Gemini-2.0-flash.

## 2.2 Evaluation of Structured Output Predictions

Irugalbandara and Li et al. evaluate their approaches using aggregated single variable measures like (micro-averaged) *precision*, *recall*, and *F1* scores. Additionally, Irugalbandara incorporates the *Geometric Mean Score* (GMS) and *Consistency* to provide a further assessment. In the work of Wang et al., *Schema Accuracy* and *Content Similarity* serve as evaluation metrics. The previously discussed Visual RAG Pipeline is assessed based on the accuracy of predictions and Ground Truth (GT) values (Lamm & Keuper, b).

Chen et al. introduce a framework that pools multiple evaluation metrics for SOPs. The responses are represented *"as objects of certain data types"*, and the metrics are calculated by *"matching of common substructures, possibly followed by normalization"* (Chen et al.). Lu et al. focus on the evaluation of LLMs output in JSON format. The analysis refers to the performance of *"valid JSON outputs against a given schema"* (Lu et al.). The following aspects are examined: *"structure understanding, escaping, and natural language description, to determine how to assess and enable LLMs to generate valid responses"* (Lu et al.). The requirements on the evaluation metrics for the proposed *mSOP-765k* benchmark has to be flexible in different ways. First, the occurrence of key items varies depending on the input image, i.e., a value to each key is not always presented in an image. Furthermore, the evaluation metric has to support the evaluation of different data types. For keys with numerical data types, a simple equality evaluation can be suitable. However, evaluation of string keys requires a semantic distance metric. Moreover, we are not only interested in the sum of the single errors over the different keys of the SOPs. We are also interested in the correctness of the entire structure of the output predictions. Consequently, we introduce combined evaluation metrics in Section 4.1.

## 2.3 Benchmarks in Related Studies

Existing structured output benchmarks differ with respect to the prediction output formats. These include formats such as *JSON* schemata and related textual structured representations, as well as visual outputs. Lu et al. provide the *SchemaBench* benchmark that consists of about 40k different *JSON* schemata. Furthermore, the *JSONSchemaBench* benchmark has been introduced by Geng et al.. This benchmark comprises about 10k *JSON* schemata.

Tang et al. introduce the *STRUC-BENCH* benchmark, which aims to evaluate the generation of structured tables from textual input. Specifically, the benchmark focuses on producing raw text tables, HTML tables, and LaTeX tables. The *StructEval* benchmark is divided into two subsets, *StructEval-T* and *StructEval-V*, designed to generate SOPs for generation and conversion tasks (Yang et al.). *StructEval-T* targets the generation of text-based structures. The supported formats within this subset include *JSON, XML, YAML,* and *Markdown.* In contrast, *StructEval-V* is dedicated to producing executable code as SOPs, enabling visual rendering of generated structures. Example output formats in *StructEval-V* encompass *HTML, Matplotlib, Canvas, LaTeX, SVG,* and more.

The previously established benchmarks have been exclusively based on textual inputs, typically formatted as JSON schemata or other text-based representations. Consequently, these benchmarks are not suitable for evaluating multi-modal inputs that combine both image and text modalities. To address this limitation, we introduce the novel *mSOP-765k* benchmark. The characterizations of the proposed benchmark are a large number of 765*k* data samples with manually annotated GT structured prediction targets. The structured output data demonstrates significant variability in multiple dimensions: it encompasses a substantial number of variables with strong variability different data types which range from simple numerical and string to complex list data types. The evaluation of numerical variables can be achieved by means of a straightforward numerical evaluation of the predictions. However, for variables with other data types, a more complex evaluation is required. Furthermore, the *mSOP-765k* benchmark combines the SOPs task with the fine-grained visual object problem due to its large-scale visual dataset. Consequently, the benchmark underscores the elevated practical relevance that may be applicable to similar real-world problems.

## 3 Dataset

The objective of this paper is to establish a benchmark for the evaluation of Structured Output Predictions from multi-modal input data. Hence, the key components for the task are the multi-modality of input data and the provision of output data in a structured format. In addition, the data types of the structured output are expected to vary to represent the complexity of real-world problems.

In the retail domain, product advertisements in leaflets exhibit the characterizations described previously. The visual product advertisement images contain a wealth of textual information, whereby the multi-modality is ensured. Information about products and promotions is typically advertised with a variety of descriptive details, including the product name, the brand, and the sales price. This output data is suitable for the provision in a structured format. Furthermore, the data types of the product and promotion data vary, from strings, integers, floating-point numbers, lists, and other data formats. The proposed large-scale *mSOP-765k* benchmark measures the ability of models to map such advertisement images to the according structured product information.

**Image Dataset:** The raw image dataset is based on the *Retail-786k* dataset (Lamm & Keuper, a). This image dataset consists exclusively of image advertisements that have been cropped from leaflet pages. It has been published in context of a fine-grained image classification benchmark in which potentially

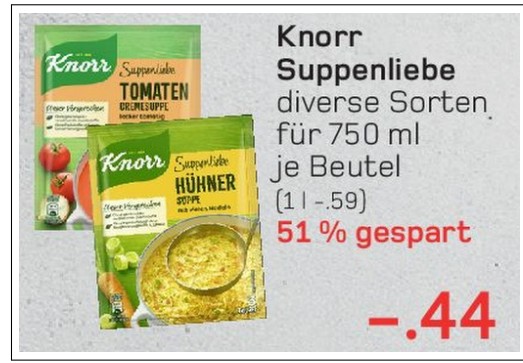

(a) Input sample: product visualizations and textual meta-information are combined in a single image.

| Data | Target | Value |
|---|---|---|
| product | brand | Knorr, Suppenliebe |
| | product category | Suppentopf, Suppen, Suppe |
| | GTINs | [08710908937095, 08710908937590] |
| | product weight | NaN |
| | different types | yes |
| promotion | price | 0.44 |
| | regular price | NaN |
| | relative discount | 51 |
| | absolute discount | NaN |

(b) Associated structured target data; missing target values are indicated by NaN. Because the inputs displays more than one product, the *GTINs* target variable holds a list of multiple product ids.

Figure 2: Representative sample from the *mSOP-765k* dataset, consisting of a query (input) image (2a) and the corresponding structured product target (2b).

visually similar products are classified by their Global Trade Item Number (GTIN) (AISBL). GTINs are internationally standardized 14-digit numbers for unique product identification. We extend the *Retail-786k* image dataset by adding structured data annotations (as described in the next paragraph) to each advertisement image. Due to data cleaning procedures based on missing structured data, the number of images in our benchmark is lower than the original 786k. In total, our provided image dataset now contains 765,463 images divided into a train split of 728,892 images and a test split of 36,571 images. The partitioning of the dataset into training and test split was adopted from (Lamm & Keuper, a). The training/test splits contain a minimum number of 10/3 images per label, where labels are defined by the semantic grouping of GTINs. In addition, our image dataset is available in two versions: one with images resized so that the longer edge measures 512 pixels, and another where the longer edge measures 256 pixels. Moreover, the image dataset exhibits a long-tail distribution with regard to the GTIN classes which is illustrated in Figure 7 in the Appendix.

**Structured Data:** In our *mSOP-765k* dataset, structured data refers to product and promotion data shown in the images. The product data comprises properties about the product(s) advertised. This data includes the information of *brand, product category, GTINs, product weight,* and *different types.* In contrast, the promotion data comprises the information on the advertised *price, regular price, relative discount,* and *absolute discount.* Most variables can be extracted directly from text in the advertisement

images, such as *price*, *discount*, and *brand*. However, other targets must be classified from the images, as *GTINs* or *product category*. Therefore, the fine-grained problem of the original dataset still exists. An example of the remaining fine-grained task is shown in Figure 3. Two images from the *mSOP-765k* dataset (3a, 3b) and the corresponding product and promotion targets per image are depicted. The fine-grained difference is illustrated by the similar visual packaging of the products and the minimal number of targets that differ in their target values (see *GTINs* and *product weight*). Other characteristics of the images in the dataset are that not all variables always have to appear in the images and that there may be several products depicted a single image. If multiple product images are displayed, this will result in a list of GTINs in the output data structure. For each image in the image dataset, the structured product and promotion have been recorded manually. These data are released in `.parquet` format for the training and test split, respectively. Table 1 presents the number of GT values for each product and promotion data in the test split. The values are reported in absolute numbers (n) and in percentages (%). Furthermore, the currency of the promotion data *price*, *regular price*, and *absolute discount* is declared in Euro (€). The promotion data *relative discount* is specified as a percentage.

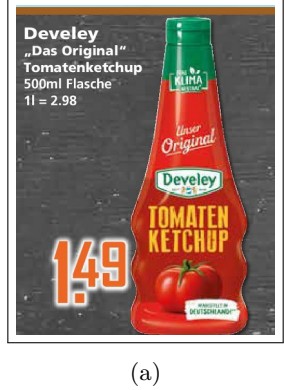 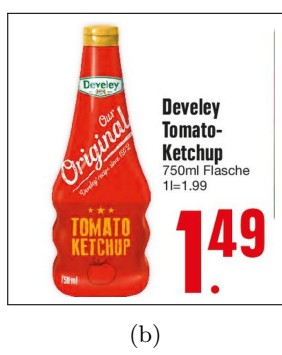

| Target | Image 3a | Image 3b |
|---|---|---|
| brand | Develey | Develey |
| product category | Tomato Ketchup | Tomato Ketchup |
| GTINs | [04006824003551] | [04006824003612] |
| product weight | 500.0 Milliliter | 750.0 Milliliter |
| different types | NaN | NaN |
| price | 1.49 | 1.49 |
| regular price | NaN | NaN |
| relative discount | NaN | NaN |
| absolute discount | NaN | NaN |

(a)           (b)

Figure 3: Illustration of the product and promoted data from the images 3a and 3b. The fine-grained data differences of the two advertisements are underlined.

Table 1: Absolute numbers (n) and percentages (%) of GT values for each target in the test split.

|  | brand | GTINs | product category | different types | price | product weight | regular price | relative discount | absolute discount |
|---|---|---|---|---|---|---|---|---|---|
| n | 36,571 | 36,571 | 36,571 | 36,571 | 36,541 | 35,365 | 12,854 | 11,719 | 667 |
| % | 100.0 | 100.0 | 100.0 | 100.0 | 99.92 | 96.70 | 35.15 | 32.04 | 1.82 |

**Representative Sample:** Figure 2 illustrates a representative sample from the *mSOP-765k* dataset. The sample consists of an advertisement image (2a) and the corresponding product and promotion data (2b). The data types of the product and promotion data vary, encompassing numerical data types (e.g., integers and floating-point numbers) as well as complex data types like lists. In this example, the product data *GTINs* are annotated as a list because the advertisement image displays more than one product and the hint of various types (ger: *diverse Sorten*) is legible. From these occurrences, the price validity of the advertisement for multiple products can be deduced.

**Data Access:** The image dataset and the structured product and promotion data can be accessed at the following URL: `https://huggingface.co/datasets/retail-product-promotion/mSOP-765k` and are licensed under a Creative Commons Attribution-NonCommercial-NoDerivatives 4.0 International License (CC BY-NC-ND 4.0) (Creative Commons). As a further addition, the description texts appearing in the advertisement images are also provided in the repository. The description texts have been extracted using the PaddleOCR tool (PaddlePaddle; Du et al.).

## 4 Evaluation

In this section, we provide a baseline empirical evaluation of current state-of-the-art SOPs methods on the proposed *mSOP-765k* benchmark. These experiments include VLM based approaches (Section 4.2) as well as a visual RAG pipeline (Section 4.3). Moreover, investigations of image and text classification on the dataset are described in Appendix A.3.

**Prompts and Structured Output Schemata:** We fix the required *system-* and *user/human-prompts* for all methods to the same input.

The *system prompt* is specified as: `You are an assistant for question-answering tasks.` While the *user/human prompt* is defined as: `Do the user-provided task on the input image. The answer must be provided in JSON format. The task is: "Extract the features.". If there is no information of a target, return NaN.`

Table 9 in the Appendix shows, for each target, whether a prediction is required and specifies the data type. Data types vary from `string`, `float`, `list of strings`, and `enumeration`. It is important to note that the product attribute *product weight* is divided into two separate queries: one for the *weight number* and one for the *weight unit*. The data types for the targets *weight unit* and *different types* are an `enumeration`. For the target *weight unit*, the valid values are: `Gramm`, `Kilogramm`, `Milliliter`, `Liter`, `Waschladungen`, `Blatt`, and `Stück` (engl. gram, kilogram, milliliter, liter, wash loads, sheet, piece). For the target variable *different types*, the possible values are: `yes` and `no`.

Table 2: Detailed description of the evaluation metrics for individual target variables within the prediction structure.

| Target | Description of Evaluation Metric |
|---|---|
| brand | ***1. String preprocessing on prediction and GT values:*** converting all characters to lowercase; removing accents and other diacritical marks using Unicode normalization; and removing all apostrophes from the strings. 
 ***2. Matching criteria:*** the Levenshtein (Lcvenshtcin) distance between the prediction and GT strings is calculated to derive a similarity score. If the similarity exceeds 0.5, the strings are considered as a match. If not, the algorithm checks whether the two strings share any common words after splitting by non-word characters. If they do, the strings are still considered as a match. 
 ***3. Accuracy calculation:*** the accuracy is computed as the proportion of matched results. |
| product weight | ***1. Concatenation:*** the prediction value is formed by combining the *weight number* and *weight unit* predictions. 
 ***2. Matching criteria:*** if the prediction string is exactly equal to the GT string, it is considered a match. If not, equivalent representations with different units (e.g., 1000g = 1kg) are treated as matches by converting between grams/kilograms or milliliters/liters based on their numerical values, followed by string equality checks. 
 ***3. Accuracy calculation:*** the accuracy is computed as the proportion of matched results. |
| different types | ***1. Accuracy calculation:*** the custom accuracy is measured based on exact string equality between the predictions and GT values which can either `yes` or `no`. |
| price data[1] | ***1. String preprocessing on prediction:*** any occurrences of the character `"-"` are removed from the prediction value (if present). 
 ***2. Accuracy calculation:*** the custom accuracy is measured based on the exact string equality between the adapted prediction and the GT value. |

---

[1]Includes the targets: *price*, *regular price*, *relative discount*, and *absolute discount*.

### 4.1 Evaluation Metrics

**Evaluation metrics for single target variables:** For all product and promotion sub-target within the structured output target, a custom accuracy score is determined. The evaluation metric is defined to ensure comparability between the predictions and the process by which the GT was generated. The GT values were specifically defined by human annotators. The metric for promotion data as well as the target *product weight* must be strict, as a subsequent comparison of identical products would otherwise not be guaranteed. For the target *brand*, an exact match is not necessary because the spellings may vary depending on the retailer's leaflets. Based on these considerations, we introduce a custom accuracy metric based on the Levenshtein distance (Lcvenshtcin), avoiding the use of an exact match calculation. The process used to calculate the score for each target is described in Table 2. Since not all targets are advertised in every image, an absence of prediction of the targets can be valid. This applies in particular to the targets *regular price*, *relative discount*, and *absolute discount* but may also occur for the other targets.

**Evaluation metrics for the structured output:** In addition to the evaluation for the single target variables, we propose a novel evaluation metric for the structured output. The measurement criteria is based on the union of the evaluation metric for single target variables. There are two distinct metrics: $\bigcup_{\text{targets}}$ and $\bigcup_{\text{test}}$. The evaluation metric $\bigcup_{\text{targets}}$ only includes samples for which the GT is given and a prediction has been delivered with regard to the considered targets. Consequently, the size of the reference set varies depending on how many target variables are considered. The evaluation $\bigcup_{\text{test}}$ also considers samples that receive a prediction but no GT is given, or where a GT is available but no prediction has been generated. In this case, the reference set is the number of the test dataset, i.e. 36,571 samples.

In general, the following applies to the evaluations in Sections 4.2 and 4.3: the experiments are executed on the *mSOP-765k* dataset with an image resolution of 512 pixels and the best result in each evaluation table is highlighted in **bold**, while the second-best result is underlined.

### 4.2 Evaluation of VLM Based Zero-Shot Approaches

We evaluated the performance of both commercial and open-source models in order to capture model diversity. All VLMs have been used in a zero-shot setting, i.e. neither training nor fine-tuning of the models has been performed on the dataset. We evaluated the following models: GPT-4o-mini in the version dated 07/18/2024 (gpt-4o-mini-2024-07-18) (OpenAI, a), Gemini-2.0-flash (gemini-2.0-flash) (Mallick & Kilpatrick, a), Gemini-2.5-flash (gemini-2.5-flash) (Mallick & Kilpatrick, b), GPT-5-mini in the version dated 08/07/2025 (gpt-5-mini-2025-08-07) (OpenAI, c;d), LLaVA version 1.6 with 34b parameters (llava-34b) (Liu et al., a;b), Qwen2.5-VL with 32b parameters (qwen2.5vl-32b) (Bai et al.), Mistral Small 3.1 with 24b parameters (mistral-small3.1-24b) (MistralAI), and Llama 3.2 Vision with 11b parameters (llama3.2-vision-11b) (Meta). The models llava-34b, qwen2.5vl-32b, mistral-small3.1-24b, and llama3.2-vision-11b have been deployed via the Ollama framework (Ollama).

Table 3 reports the percentage of evaluable predictions for the models on the test split of *mSOP-765k* dataset introduced in Section 3. The numerical values in the table indicate the percentage of predictions that can be evaluated. A prediction is considered evaluable only if the corresponding predictions for all targets - except the targets *product category* and *GTINs* - are defined, i.e. no missing entries are allowed. Note that the prediction values of not advertised product and promotion data are indicated by `NaN` which is accepted as a valid value. The highest rate of evaluable responses has been achieved by the VLM gpt-5-mini-2025-08-07. The VLM gpt-4o-mini-2024-07-18 has returned only eight invalid predictions, corresponding to 99.98% appraisable responses. The VLMs gemini-2.5-flash and gemini-2.0-flash have achieved 99.95% and 99.56% reviewable predictions. The VLMs llava-34b, qwen2.5vl-32b, and mistral-small3.1-24b have provided a similar magnitude of evaluable predictions whereas the VLM llama3.2-vision-11b has returned 78.95% assessable predictions.

The prediction of each product and promotion target variable is not feasible using VLMs alone. Specifically, information about the targets' *product category* and *GTINs* is not presented in the advertisement images. Consequently, no valid predictions can have been generated for these targets and, therefore, no evaluations have been conducted in this zero-shot setting.

Table 3: Correctness of output structures: Numerical values represent the percentage of evaluable predictions, for the test split and for each VLM. Predictions are considered to be evaluable if the output structures contain all requested variables with the correct type. Note: positive evaluability of output structures does not implicate correctness of the prediction.

| gpt-4o-mini-2024-07-18 | gemini-2.0-flash | llava-34b | qwen2.5 vl-32b | mistral-small3.1-24b | llama3.2-vision-11b | gemini-2.5-flash | gpt-5-mini-2025-08-07 |
|---|---|---|---|---|---|---|---|
| 99.98% | 99.56% | 99.12% | 99.54% | 96.53% | 78.95% | 99.95% | **100.00%** |

Table 4: VLM-based – Scores of the evaluation metrics for each single target variable and VLM. Percentages are calculated based on comparable GT and predicted values.

| | gpt-4o-mini-2024-07-18 | gemini-2.0-flash | llava-34b | qwen2.5 vl-32b | mistral-small3.1-24b | llama3.2-vision-11b | gemini-2.5-flash | gpt-5-mini-2025-08-07 |
|---|---|---|---|---|---|---|---|---|
| brand | 95.55% | 96.61% | 70.79% | 93.37% | 94.12% | 90.77% | **96.73%** | 95.82% |
| product weight | 71.26% | **80.97%** | 8.04% | 17.12% | 16.71% | 15.76% | 72.44% | 80.49% |
| different types | 85.48% | 88.97% | 60.23% | 75.64% | 79.62% | 56.03% | 89.99% | **90.70%** |
| price | 97.19% | 97.37% | 52.53% | 94.15% | 94.28% | 91.57% | 97.37% | **97.53%** |
| regular price | 85.93% | 92.47% | 19.00% | 65.19% | 84.14% | 51.79% | **94.89%** | 92.84% |
| relative discount | 98.75% | **99.02%** | 48.40% | 97.90% | 96.19% | 92.68% | 98.93% | 98.83% |
| absolute discount | 82.01% | **92.63%** | 11.41% | 72.07% | 62.19% | 40.11% | 91.89% | 86.51% |

Figure 13 in the Appendix illustrates the difference between the number of predictions and the number of GT values for all target variables - except *product category* and *GTINs* - and VLMs. For the targets *brand*, *product weight*, *different types* and *price*, the number of predictions has matched the number of GT values for all VLMs closely, except for the llama3.2-vision-11b model. Other models have produced slightly more or fewer predictions. A notable disparity among the VLMs is evident in the number of predictions for the target *regular price*. The VLMs qwen2.5vl-32b and llama3.2-vision-11b clearly have generated fewer predictions than the other investigated models.

Figure 15 in the Appendix presents the prediction error in € for the target variable *price* and each VLM. The prediction error is defined as the difference obtained by subtracting the predicted value from the GT value. The VLMs llava-34b and qwen2.5vl-32b have demonstrated notable outliers, with deviations reaching up to -35,000€. Evaluations of the other models reveal a narrower distribution; however, outliers of considerable magnitude, approximately -4,000€, are also present. These findings are illustrated in Figure 14 in the Appendix. Moreover, the median of each VLM evaluation is 0.00€.

Table 4 summarizes the results of the evaluation metrics for single target variables. The percentages refer to the subset of comparable GT values and predictions, i.e. only entries with defined values are considered. The VLMs gemini-2.0-flash, gemini-2.5-flash and gpt-5-mini-2025-08-07 consistently have achieved the best results. Among all product and promotion data, the VLM gemini-2.5-flash has provided best or second best accurate predictions. The VLM gemini-2.0-flash has delivered the highest custom accuracy score of 99.02% for the target variable *relative discount*.

Table 14 in the Appendix shows the results of the evaluation for structured output when the target variables are systematically incorporated based on the set of comparable GT values and predictions, i.e. only entries with defined values are considered. Hence, the evaluation metric $\bigcup_{\text{targets}}$ is calculated. Initially, the score for the target variable *brand* is shown; subsequently, the score of the union of the target variables *brand* and *product weight* are considered, and so on. Typically, the score decreases as the union of the variables under consideration expands. This incidence holds for nearly all models. Exceptions have occurred in the evaluations of the VLMs llava-34b and qwen2.5vl-32b. For example, for the VLM llava-34b, adding the target

Table 5: VLM-based – Results of the evaluation metric $\bigcup_{\text{test}}$ for the structured output. Scores are calculated based on comparable GT and predictions values relative to the entire test dataset.

| | gpt-4o-mini-2024-07-18 | gemini-2.0-flash | llava-34b | qwen2.5 vl-32b | mistral-small3.1-24b | llama3.2-vision-11b | gemini-2.5-flash | gpt-5-mini-2025-08-07 |
|---|---|---|---|---|---|---|---|---|
| brand | 95.53% | 96.19% | 70.17% | 92.94% | 90.85% | 71.66% | **96.68%** | 95.82% |
| $\bigcup_{\text{test}}$ product weight | 66.08% | **75.81%** | 5.54% | 15.51% | 15.30% | 14.14% | 68.02% | 74.84% |
| $\bigcup_{\text{test}}$ different types | 24.33% | 30.58% | 0.87% | 3.04% | 4.51% | 5.23% | 29.53% | **32.57%** |
| $\bigcup_{\text{test}}$ price | 23.50% | 29.65% | 0.46% | 2.88% | 4.29% | 4.93% | 28.65% | **31.67%** |
| $\bigcup_{\text{test}}$ regular price | 6.83% | 9.59% | 0.04% | 0.04% | 0.95% | 0.01% | 9.67% | **10.32%** |
| $\bigcup_{\text{test}}$ relative discount | 3.66% | 4.87% | 0.01% | 0.01% | 0.44% | 0% | 4.92% | **5.28%** |
| $\bigcup_{\text{test}}$ absolute discount | 0% | 0% | 0% | 0% | 0% | 0% | 0% | 0% |

*different types* to the previously union of targets, the results have improved due to a substantial reduction in the number of evaluable GT values and predictions.

Table 5 provides a comprehensive evaluation of the metric for structured output related to the entire test split of the dataset. Once again, gemini-2.0-flash, gemini-2.5-flash and gpt-5-mini-2025-08-07 have dominated the top results However, it has not been possible to predict all product and promotion targets for any single advertisement image in the test dataset. This follows from the score value of 0% for all product and promotion data; refer to the last row of the table.

Table 15 in the Appendix illustrates the average elapsed time per request in seconds, as well as the total elapsed time in hours (optionally in days) for the test split of the dataset and for each VLM. The gemini-2.0-flash model has required the least amount of time, with a total elapsed time of 21.7 hours for all requests, followed by the VLM qwen2.5vl-32b, which accumulated a total elapsed time of 46.2 hours. Furthermore, the average total costs per request and the total costs given in USD are presented. The costs for the open-source VLMs have been estimated based on the assumption of a Linux VM in Microsoft Azure (Microsoft). The minimal total costs for processing the test split requests using the VLM gemini-2.0-flash has been approximately USD 7.40, followed by the VLM qwen2.5vl-32b at around USD 8.70.

### 4.3 Evaluation of a Visual RAG Based Approach

For the evaluation of the SOPs generated by a visual RAG based approach, we focused on the Visual RAG Pipeline (Lamm & Keuper, b). Figure 4 represents the architecture of the Visual RAG Pipeline. The pipeline consists of five main steps: *Preprocessing, Vector Store, Retrieval / Classification / Relational Query, Prompt Generation / Completion*. The pipeline is based on the RAG approach (Gao et al.) and is distinguished by an extension of the prompt using historical context. The context comprises historical samples that are similar to the input image. These historical samples consist of advertisement images along with their corresponding product and promotion data. In the last step, called *Completion*, a VLM request is applied. Therefore, we investigated the following VLMs: gpt-4o-mini-2024-07-18, gemini-2.0-flash, gemini-2.5-flash, qwen2.5vl-32b, and mistral-small3.1-24b. The commercial VLMs mentioned were selected taking into account both time constraints and cost efficiency. The VLM qwen2.5vl-32b has needed the lowest elapsed time per request, as well as the lowest total costs of all requests with regard to the open-source models (Table 15 in the Appendix). As another open-source model, the VLM mistral-small3.1-24b were examined.

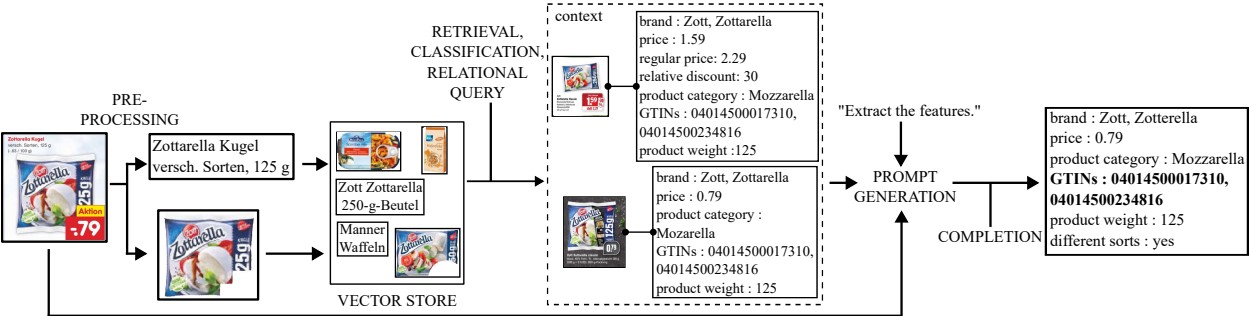

Figure 4: Illustration of the architecture of the Visual RAG Pipeline as presented in (Lamm & Keuper, b). The pipeline consists of five main steps and is distinguished by an extension of the prompt using historical context and the RAG approach (Gao et al.).

Table 6 shows the percentage of evaluable predictions. The VLMs gpt-4o-mini-2024-07-18, gemini-2.5-flash, and mistral-small3.1-24b have achieved a maximal evaluability of 98.14%. The VLM qwen2.5vl-32b has providedappraisable predictions for 11 promotion images more than the gemini-2.0-flash model.

The difference between the number of predictions and the number of GT values are presented in Figure 17 in the Appendix. For all product variables and the promotion variable *price*, there have been minimal fewer predictions than the available GT values. This results from the fact that the number of fewer predictions is the number of requests that have returned non-defined values for all targets variables. For the remaining target variables, all models have produced a higher number of predictions, the VLMs gpt-4o-mini-2024-07-18 and gemini-2.0-flash have generated particularly between 20,000 and 25,000 additional predictions.

Table 6: Correctness of output structures: Numerical values represent the accuracy of evaluable predictions, expressed as percentages, for the test split obtained by the Visual RAG Pipeline for each VLM. Invalidity mainly arises from errors or request timeouts.

| gpt-4o-mini-2024-07-18 | gemini-2.0-flash | qwen2.5 vl-32b | mistral-small3.1-24b | gemini-2.5-flash |
|---|---|---|---|---|
| **98.14%** | 98.03% | 98.06% | **98.14%** | **98.14%** |

In Figure 16 in the Appendix, the prediction errors for the target *price* per VLM in the Visual RAG Pipeline is analyzed. The VLM gpt-4o-mini-2024-07-18 has exhibited the most pronounced outliers in both positive and negative directions, exceeding 150.00€ and dropping below -150.00€. The distribution of the VLMs gemini-2.0-flash, qwen2.5vl-32b, and mistral-small3.1-24b are very similar. Consistently, the median value for each VLM evaluation have remained 0.00€.

Table 7: Visual RAG-based – Scores of the evaluation metrics for each single target variables and VLM used in the Visual RAG Pipeline. Percentages are calculated based on comparable GT and predicted values.

|  | gpt-4o-mini-2024-07-18 | gemini-2.0-flash | qwen2.5 vl-32b | mistral-small3.1-24b | gemini-2.5-flash |
|---|---|---|---|---|---|
| brand | 98.53% | 95.21% | 90.04% | 90.01% | **98.81%** |
| product category | **83.30%** | 82.54% | 80.26% | 78.76% | 70.58% |
| GTINs | 70.87% | 71.28% | 69.26% | **71.56%** | 66.85% |
| product weight | 85.68% | **85.96%** | 80.72% | 80.63% | 85.75% |
| different types | **91.55%** | 89.61% | 88.41% | 79.95% | 88.64% |
| price | **96.48%** | 59.07% | 18.57% | 18.52% | 96.45% |
| regular price | 84.71% | 68.90% | 26.90% | 27.33% | **93.76%** |
| relative discount | 95.99% | 70.39% | 17.72% | 15.98% | **97.98%** |
| absolute discount | 77.37% | 70.80% | 25.24% | 20.78% | **90.52%** |

Table 7 shows the results of the evaluation metric for the single target variables. The VLMs gpt-4o-mini-2024-07-18 and gemini-2.5-flash have delivered the best and second-best scores for all target variables except *GTINs* and *product weight*. For the variable *GTINs*, the VLM mistral-small3.1-24b has achieved the highest score with 71.56% and for the target *product weight*, the VLM gemini-2.0-flash has attained 85.96%.

The evaluation metric for the structured output in relation to the valid GT and prediction values, i.e. $\bigcup_{targets}$ is calculated, is presented in Table 20 in the Appendix. The VLMs gpt-4o-mini-2024-07-18 and gemini-2.0-flash have dominated the results by always achieving the best and the second-best scores, respectively. Although these models have outperformed the results, the evaluation of the scores in relation to the test split, i.e. $\bigcup_{test}$ is calculated, shows weakness. Table 8 demonstrates that although the two models continue to dominate the results, the Visual RAG Pipeline likewise have failed to produce correct predictions for all product and promotion variables within any single promotion image.

Table 21 in the Appendix shows the evaluation of the elapsed time and the total costs per request and for all requests using the Visual RAG Pipeline. The shortest duration for the total elapsed time for all requests has been achieved by the VLM gemini-2.0-flash, with a duration of 42.04 hours. In relation to the total costs for all requests, the VLMs mistral-small3.1-24b and qwen2.5vl-32b have been significantly cheaper than the other models. It should be noted that the cost estimates for the open-source VLMs have been based on running a Linux VM in Microsoft Azure (Microsoft). Among these, the previously named models have stood out with estimated costs of approximately USD 13.90 and USD 17.10, respectively. Crucially, the costs only represent the step *Completion* of the Visual RAG Pipeline. In this evaluation, additional costs from the other pipeline steps were not considered.

Table 8: Visual RAG-based – Results of the evaluation metric $\bigcup_{test}$ for the structured output. Scores are calculated based on comparable GT and predictions values relative to the entire test dataset.

| | gpt-4o-mini-2024-07-18 | gemini-2.0-flash | qwen2.5vl-32b | mistral-small3.1-24b | gemini-2.5-flash |
|---|---|---|---|---|---|
| brand | 96.70% | 93.33% | 88.29% | 88.41% | **96.98%** |
| $\bigcup_{test}$ product weight | 81.92% | 81.41% | 75.59% | 75.50% | **82.14%** |
| $\bigcup_{test}$ different types | 39.00% | 39.00% | 35.68% | 35.98% | **39.73%** |
| $\bigcup_{test}$ price | 37.55% | 25.74% | 8.20% | 8.30% | **38.01%** |
| $\bigcup_{test}$ regular price | 11.17% | 8.70% | 0.68% | 0.94% | **12.56%** |
| $\bigcup_{test}$ relative discount | 5.97% | 4.40% | 0.30% | 0.40% | **6.58%** |
| $\bigcup_{test}$ product category | **5.36%** | 3.92% | 0.27% | 0.38% | 5.03% |
| $\bigcup_{test}$ GTINs | **4.78%** | 3.49% | 0.24% | 0.35% | 4.49% |
| $\bigcup_{test}$ absolute discount | 0% | 0% | 0% | 0% | 0% |

### 4.4 Error Analysis of VLM Based and Visual RAG Based Approaches

**Validity of Predictions Analysis.** The validity of the predictions has been reduced using the Visual RAG Pipeline (see: Tables 3 and 6). For instance, the VLM gpt-4o-mini-2024-07-18 has achieved 99.98% valid predictions by using VLM based approach in contrast to using the Visual RAG Pipeline with 98.14%. The drop in valid predictions, or rather undefined values, from the requests originates from issues in the previous steps in the Visual RAG Pipeline. Hence, the number of valid predictions from the pipeline depends on multiple factors in the different steps, and not only on the VLM used in the step *Completion*.

**Analysis of Evaluation Metrics for Single Variables.** Tables 4 and 7 show the scores of the evaluation metrics for each single target variable. For the variable *price*, there has been a significant difference between the evaluation of the two investigated approaches. The VLM based approach using gemini-2.0-flash has

achieved a score of 97.37% while the visual RAG based approach has only reached 59.07%. This difference is approximately 40 percentage points. The false predictions using Visual RAG Pipeline have been followed by the fact that the predictions have been identical to the price information of the historical data provided in the VLM prompt. The percentage of price predictions identical to historical price information for the VLM gemini-2.0-flash used in the Visual RAG Pipeline has been 66.75%. The evaluations of the target *price* for the VLM gemini-2.5-flash do not demonstrate such behavior. In this instance, a discrepancy of about 1 percentage point has been found between the evaluation of the VLM based and Visual RAG based approach.

**Analysis of Evaluation Metrics for Structured Output.** The results of the evaluation metric $\bigcup_{\text{targets}}$ are illustrated in Tables 14 and 20 in the Appendix. It can be deduced from these tables (when taking into account the target variable *brand* and the VLMs gemini-2.5-flash and gpt-4o-mini-2024-07-18) that the score using the Visual RAG Pipeline has been about 2 and 3 percentage points higher than using the VLM alone. All other models have returned worse scores using the Visual RAG Pipeline for the same variable. For the union of the target variables *brand* and *product weight*, it has been generated higher scores from all VLMs used in the visual RAG based approach. The integration of additional variables has led to a reduction in scores. Moreover, the table comparison shows that, when using the VLMs alone, only the gemini-2.5-flash model has produced better results comparable to the Visual RAG based approach. The evaluations of the right predictions in relation to the entire test dataset are illustrated in Tables 5 and 8. The scores using Visual RAG Pipeline with the best VLM (gemini-2.5-flash) have been reached in higher percentages than using the best models (gemini-2.5-flash and gpt-5-mini-2025-08-07) in the VLM based approach in terms of the union of variables from *brand* up to *relative discount* inclusively. It is also important to mention that *product category* and *GTINs* is only predictable by the visual RAG based approach. This characterization stands out for this method. Consequently, the analysis of the two tables also shows that there has been no promotion image for which all the predictions of the target variables are valid.

**Qualitative Error Analysis.** In addition, an analysis of three example promotion images is presented. Figure 5 displays the promotion images and the corresponding approach combination that has correctly predicted the target variables (excluded: *absolute discount*), whereas the other approach has failed. Moreover, Figure 5c shows a promotion image for which both methods have made incorrect predictions. For the promotion image in Figure 5a, the VLM gemini-2.0-flash has correctly predicted (all) product and promotion data. It is important to mention that using the VLM based approach, the product data *GTINs* and *product category* have not been predicted because these information are not printed in the advertisement image. The predictions for the promotion image shown in Figure 5a are listed in Table 22 in the Appendix. The false predictions of the Visual RAG based approach Visual RAG Pipeline/gpt-4o-mini-2024-07-18 have been caused by the information used from historical data. For the advertisement image shown in Figure 5b, the list of predictions is provided in Table 23 in the Appendix. The prediction using the VLM gemini-2.0-flash for the target *product weight* (consists of requests for *weight number* and *weight unit*) has differed to the GT values. In contrast to this, the method Visual RAG Pipeline/gpt-4o-mini-2024-07-18 has predicted all targets

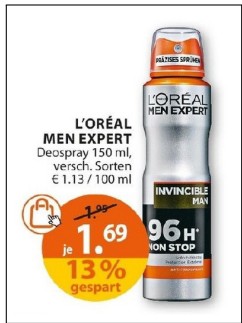 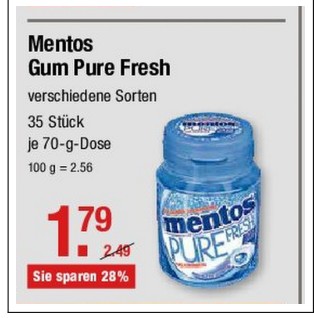 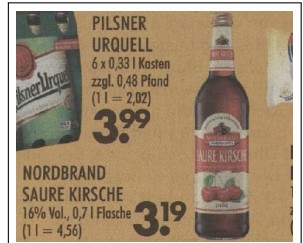

(a) VLM/gemini-2.0-flash     (b) Visual RAG Pipeline/gpt-4o-mini-2024-07-18     (c) neither of the two methods

Figure 5: Illustration of promotion images and the approach combination that generates correct SOPs. The predictions and GT values have been returned from the approaches are shown in Tables 22 to 24 in the Appendix.

correctly (exclusively: *absolute discount*). The prediction listings for the advertisement image, illustrated in Figure 5c, are shown in Table 24 in the Appendix. The incorrect predictions have likely been caused by the fact that two products are visible in the advertisement image, along with their product and promotion data.

**Error Analysis With Minimal Training Images.** As a further qualitative analysis, we evaluated the evaluation metric for the structured output for a label containing only the minimal number of training images (10 images) and test images (3 images). One of the promotion test images and the predictions of the two approaches VLM/gemini-2.0-flash and Visual RAG Pipeline/gpt-4o-mini-2024-07-18 are shown in Figure 6 and Table 25 in the Appendix. Both approaches have correctly predicted the targets *brand*, *product weight*, and *price*. However, for the other targets, incorrect predictions have been observed. This discrepancy has arisen because the GT values do not always align with the information displayed in the images. For instance, some images lack any indication of different types, although the GT values specify them. Furthermore, both approaches occasionally have generated predictions for targets that are not visually present in the images, particularly for *regular price*, *relative discount*, and *absolute discount*.

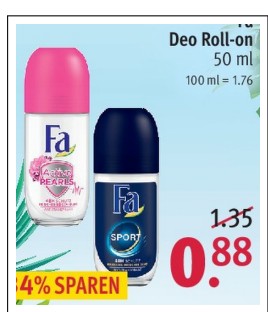

Figure 6: One of three test images from a label with only 10 training images.

## 5 Discussion, Limitations and Outlook

In this paper, we present the *mSOP-765k* benchmark that comprises about 765k samples, primarily product advertisement images alongside the corresponding product and promotion data. These data are characterized by their structured format with data types varying depending on the specific variable. Moreover, not all images necessarily feature the same set of variables, as the elements being advertised can differ from one image to another. Our evaluations of SOPs focus on VLM based approaches as well as a visual RAG based approach. To evaluate the different methods, self-defined metrics for single variables and structured output are developed and investigated.

The evaluation of the Visual RAG Pipeline indicated that for certain variables, such as *GTINs* and *product category* - which are not depicted in promotion images - training data like historical data are required. However, for other targets, this approach can be unsuccessful, as it can lead to erroneous predictions resulting from the direct adoption of historical data, especially for the promotion data. Figure 18a in the Appendix shows an advertisement image for which the Visual RAG based approach incorrectly predicts the target *price*. The false predictions are enumerated in Figure 18b in the Appendix. Furthermore, the historical data of the target *price* is presented that were incorporated into the prompt. The majority of the predictions coincide with the values found in the historical data, indicating a tendency for the model to replicate these inputs as its predictions. Similarly, the evaluations of VLMs demonstrate that accurately predicting product and promotion data through a single unified process remains a significant challenge.

The error analysis of the VLM based approach reveals that VLMs exhibit challenges in accurately interpreting images containing multiple price specifications. Detailed price prediction results for each VLM, corresponding to the advertisement image presented in Figure 19a in the Appendix, are provided in Figure 19b in the Appendix. Moreover, product weight specifications may be represented using different units of measurement. For instance, the product advertisement illustrated in Figure 20a in the Appendix includes weight information expressed both in grams and in pieces. The associated product weight predictions are documented in Figure 20b in the Appendix. Section A.8 in the Appendix shows that the fine-tuning based approach is not eligible because of its high cost and lengthy training time. Fine-tuned models are static, requiring retraining whenever the range of product advertisements change. Moreover, the evaluations show that the fine-tuned models tend to be overfitted. In conclusion, the SOPs task stays unresolved with the *mSOP-765k* dataset. Based on these findings, the choice of approach may depend on the promotion and product variable in order to achieve better results. Moreover, further investigation into reasoning models (Zhong et al.), which offer additional potential by incorporating a thought process, is warranted. Finally, the proposed *mSOP-765k* benchmark provides a solid foundation for future research and can be used as a benchmark in a variety of tasks.

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
