# OpenReview forum: "mSOP-765k: A Benchmark For Multi-Modal Structured Output Predictions"
_TMLR — Accepted by TMLR_

### Review · Reviewer_zGWs · 2025-10-07

**Summary Of Contributions:**

This work presents a large-scale benchmark for the evaluation of multimodal Structured Output Prediction. Concretely, an extension of the Retail-786k dataset, which contains real-world product advertisements, and which in this work is augmented with manual annotations for product name or brand, and numerical data such as product weight, price, and discount. It also introduces ad-hoc metrics for the evaluation of the prediction of labels in an escenario of multiclass multilabel classification.

Strengths:
- The evaluation of VLM-based models and Visual RAG-based models in itself is already a contribution.
- The work is of high relevance for the TMLR community, as it introduces the largest corpus of its kind up to now.
- The document is well written. All ideas flow smoothly, and the contributions are clearly grasped.
- The constraints for the different tasks that are evaluated are well defined.
- The analysis of results, in quantitative terms, provides good hints for future applications.

Opportunities for improvement:
- The limitations of current state-of-the art seems poorly explored or explained.
- There is a fixed split for training and test sets. Although this is often the way to make datasets public, it lacks a statistical analysis of the distribution of variables which could provide hints on whether data points are separable or not. For instance, color or intensity pixel distribution, or the distribution of outputs of hidden layers of a pre-trained CNN model, or sentence embeddings.
- It seems that Table 3 refers to recall, and Table 4 to precision. If so, please use this common terminology. Otherwise, more detail about the discrepancy between the numeric results must be provided.

Other minor observations:
- It is often better if figures and tables appear right after the paragraph where they are mentioned for the first time.
- The explanation about "Evaluation metrics for the structured output" is a bit unclear. It could be improved.
- The architecture for the visual RAG could be included in the main document.

**Audience:**

Yes

**Audience Explanation:**

The findings are of high relevance to the machine learning community.

**Broader Impact Concerns:**

No particular comment on this regard.

**Claims And Evidence:**

Yes

**Claims Explanation:**

Claims are support by either, citations to relevant work, of the results presented in the submitted manuscript.

**Requested Changes:**

It seems that Table 3 refers to recall, and Table 4 to precision. If so, please use this common terminology. Otherwise, more detail about the discrepancy between the numeric results must be provided.

---

### Review · Reviewer_6iZv · 2025-10-08

**Summary Of Contributions:**

The authors propose a dataset to test the multimodal capabilities of VLMs. This dataset is based on product advertisements, which typically display the product, with a canonical brand, and other information like the price, size, weight, or discount. This allows the target predictions to follow a structured verifiable format, making it easier to benchmark. The performance of different open-source and closed source VLMs is evaluated.

The authors claim that the proposed benchmark is a challenging problem that cannot yet be completely solved. To me, this may have been true at time of creation, when the most powerful VLMs were GPT-4o-2024-07-18 and Gemini 2.0 Flash, but the SOTA has progressed quite far since then, with the paper itself acknowledging that GPT-4o achieves a 99.98% valid prediction rate (see Table 3). Prediction rates only go down when filtered to single target attributes.

Additionally, the authors mention all images are resized to be at most 512 pixels wide. In my experience this can often make small text close to illegible, even for humans, and it's possible it may not even be possible to achieve 100% accuracy.

To me, this lowers the usefulness of the dataset, but it does seem like an interesting dataset to have around, and even datasets that are mostly-solved can have value (see a large number of papers built off of MNIST, for example)

**Audience:**

Yes

**Audience Explanation:**

Could be of interest to people working on VLMs

**Claims And Evidence:**

Yes

**Claims Explanation:**

Paper provides enough details on dataset contruction, VLMs used, scoring mechanisms, etc.

**Requested Changes:**

N/A

---

### Review · Reviewer_5YLW · 2025-11-28

**Summary Of Contributions:**

Contributions:

- Introduces mSOP-765k, a large-scale benchmark for multi-modal structured output prediction (mSOP) with 765k product advertisement images annotated with structured data (text, numbers, lists).
- Provides novel evaluation metrics for complex structured outputs, beyond single-value predictions.
- Performs a baseline evaluation using LLMs, VLMs, Vision Transformers (ViTs), and multi-modal RAG pipelines, showing current state-of-the-art methods struggle on multi-value predictions.
- Dataset and benchmark are openly available under a Creative Commons license.

Strengths:

- Scale and diversity: Large dataset with real-world images and multi-modal inputs.
- Structured output focus: Supports machine-readable outputs (key-value pairs) covering multiple data types (strings, floats, lists).
- Comprehensive evaluation: Metrics for both individual targets and full structured outputs, enabling robust benchmarking.
- Baseline analysis: Highlights current model limitations and provides reference points for improvement.

Weaknesses / Limitations:

- Zero-shot limitations: VLMs and LLMs fail to predict all targets accurately, especially multi-value outputs like GTINs and product categories.
- Complexity: Long-tail distributions and multiple products per image make accurate prediction challenging.
- Partial coverage: Not all targets are present in every image; some require manual extraction or OCR preprocessing.
- State-of-the-art gaps: Existing methods perform well on single targets but struggle with consistent multi-value structured outputs.

**Audience:**

Yes

**Audience Explanation:**

VLLM are certainly a hot topic these days.

**Broader Impact Concerns:**

would store names have an impact in your model learning/predictions?

**Claims And Evidence:**

Yes

**Claims Explanation:**

Clear benchmark scenarios.

**Requested Changes:**

- Baseline Models: Include additional baselines, particularly fine-tuned LLMs/VLMs, to allow more meaningful performance comparisons beyond zero-shot evaluation.

- Evaluation Metrics Clarity: Explain in more detail how the structured output metrics handle missing values, lists, and multi-product entries. Include step-by-step examples for clarity.

- Zero-Shot Limitations: Discuss the limitations of VLMs in zero-shot settings more explicitly, highlighting which targets remain unsolved and why.

- Error Analysis: Provide qualitative examples of prediction errors, including extreme outliers in price predictions, to illustrate real-world challenges.

- Handling Missing/NaN Values: Include clear guidelines on how missing or NaN target values are treated during evaluation and how models are expected to respond.

- Long-Tail Distribution: Discuss strategies to handle class imbalance, given the long-tail distribution of GTINs and other product categories, to help interpret results in a realistic retail setting.

- Multi-Product Images: Include analysis of images containing multiple products and how predictions of list-type attributes (e.g., GTINs) are evaluated and scored.

---

> ### Author Response · Authors · 2025-12-08
>
> The reviewer lists four weaknesses (see details below). We agree that current methods investigated in our benchmark are struggling with these issues. However, we would argue that this is generally not a weakness of our benchmark, exposing these shortcomings.
>
> **W1:** Zero-shot limitations: VLMs and LLMs fail to predict all targets accurately, especially multi-value outputs like GTINs and product categories.
>
> **A1**: Some target values like GTINs can not be predicted in a zero-shop approach because this information is not directly embedded in the test images and has to be inferred from additional context. Therefore, we intentionally split the evaluation into two parts:  a zero-shot evaluation in 4.2, and a “in context” evaluation in 4.3. The latter is evaluated by RAG approaches. We will add a fine-tuning approach as well.
>
> **W2:** Complexity: Long-tail distributions and multiple products per image make accurate prediction challenging.
>
> **A2:** We agree that long tail problems are hard, but since they are very common in real-world problems (products are advertised with varying frequency) we would argue that long-tailed benchmarks give a more realistic indication of model performance, refer to figure 8 in the appendix.
>
> **W3:** Partial coverage: Not all targets are present in every image; some require manual extraction or OCR preprocessing.
>
> **A3:** The benchmark makes the explicit assumption that models need to have text comprehension (OCR) abilities in order to solve the problem - that makes the benchmark multi-modal. Regarding the coverage: please also refer to W1.
>
> **W4:** State-of-the-art gaps: Existing methods perform well on single targets but struggle with consistent multi-value structured outputs.
>
> **A4:** Indeed, one of the findings in our evaluation is that state-of-the art models still struggle with multi-value output predictions. Take Gpt-5 for example: multiplying the single value accuracies in table 4 would result in a combined chance of ~50% for an accurate prediction, while the result for the  structured prediction shown in table 5 is 0%. This shows that for practical reasons it would be more accurate (but way more expensive) to query each target independently - leaving much room for further improvement of structured output prediction models (to be measured with our benchmark).

---

> > ### Author Response · Authors · 2025-12-08
> > **Part II**
> >
> > ***Requested changes:***
> >
> > **RC1:** Baseline Models: Include additional baselines, particularly fine-tuned LLMs/VLMs, to allow more meaningful performance comparisons beyond zero-shot evaluation.
> >
> > **A1:** We have added new baselines for GPT-5 and Gemini2.5 models. We also agree that fine-tuned models are an important approach which should be considered besides RAG. However, fine-tuning is very compute and cost intensive. Fine-Tuning runs of an OpenAI and a local model (gemma-3) are currently on the way. We expect results by the end of the week and will add an additional section discussing fine-tuning to the paper.
> >
> >
> > **RC2:** Evaluation Metrics Clarity: Explain in more detail how the structured output metrics handle missing values, lists, and multi-product entries. Include step-by-step examples for clarity.
> >
> > **A2:** We add the description of the evaluation metric process depending on missing value and list of values in section A7 in the appendix. Table 17b shows a missing price prediction for the model Llama 3.2 and table 21 presents for the GTINs a list of predictions.
> >
> > **RC3:** Zero-Shot Limitations: Discuss the limitations of VLMs in zero-shot settings more explicitly, highlighting which targets remain unsolved and why.
> >
> > **A3:** Thank you for pointing this out. We have updated the regarding sections and added additional figures 17, 18 to further clarify this.
> >
> >
> > **RC4:** Error Analysis: Provide qualitative examples of prediction errors, including extreme outliers in price predictions, to illustrate real-world challenges.
> >
> > **A4**: We have added figures 17 and 18 to show more qualitative samples and figure 13 showing outlier statistics.
> >
> > **RC5**: Handling Missing/NaN Values: Include clear guidelines on how missing or NaN target values are treated during evaluation and how models are expected to respond.
> >
> > **A5:** If a certain variable is not present in a sample, e.g. no “absolute discount” is displayed in the advertisement, the correct answer a model is expected to give is NaN. We have updated section 4.1 in the paper for clarification.
> >
> > **RC6::** Long-Tail Distribution: Discuss strategies to handle class imbalance, given the long-tail distribution of GTINs and other product categories, to help interpret results in a realistic retail setting.
> >
> > **A6:** Only two of the target variables (GTIN and Product Category) are long tailed. We agree that long-tailed classification problems are often difficult. However, we would argue that this is the realistic scenario in many practical applications. We add figure 8 to the appendix which shows a high variance for product categories in the long tail, but no striking dominance of more frequent products.
> >
> >
> > **RC7:** Multi-Product Images: Include analysis of images containing multiple products and how predictions of list-type attributes (e.g., GTINs) are evaluated and scored.
> >
> > **A7:** The predictions of list-type attributes including the target GTINs and product category are considered as correct if all elements are predicted in the same order as in the GT list.
> >
> >
> > ***Broader Impact Concerns:***
> > **BIC1:** would store names have an impact in your model learning/predictions?
> >
> > **A1**:We do not expect any impact of adding store names (at training or inference time). The benchmark has been specifically designed to measure generalization beyond implicit or explicit store/retailer biases which for example could arise from specific advertisement styles like fonts, backgrounds, layout etc. For that reason, we separate stores/retail brands in train and test sets - product advertisements from a single source (store/retailer) are only used in train OR test.

---

### Decision · Action_Editor_E8Ty · 2026-01-05

**Recommendation:** Accept as is

**Audience:**

Yes

**Audience Explanation:**

VLM is currently a hot topic and the presented dataset could be of interest to people working in this area.

**Claims And Evidence:**

Yes

**Claims Explanation:**

In this paper, the authors present a large-scale dataset to test the multi-modal capabilities of VLMs. The dataset is an extension of Retail-786k, which contains real-world product advertisements, and in this work is augmented with manual annotations for product name or brand, and numerical data such as product weight, size, price, and discount. The paper also introduces ad-hoc metrics for the evaluation of label prediction and uses them to evaluate the performance of several open- and closed-source VLMs.

The main contribution is an interesting benchmark that can be used in evaluation of VLMs, which currently have many applications.